# Plant-Derived Exosome-like Nanoparticles for Biomedical Applications and Regenerative Therapy

**DOI:** 10.3390/biomedicines11041053

**Published:** 2023-03-29

**Authors:** Andari Sarasati, Muhammad Hidayat Syahruddin, Archadian Nuryanti, Ika Dewi Ana, Anggraini Barlian, Christofora Hanny Wijaya, Diah Ratnadewi, Triati Dewi Kencana Wungu, Hiroshi Takemori

**Affiliations:** 1Doctoral Study Program, Faculty of Dentistry, Universitas Gadjah Mada, Yogyakarta 55281, Indonesia; 2Department of Dental Biomedical Sciences, Faculty of Dentistry, Universitas Gadjah Mada, Yogyakarta 55281, Indonesia; 3Research Collaboration Center for Biomedical Scaffolds, National Research and Innovation Agency of the Republic of Indonesia, Yogyakarta 55281, Indonesia; 4School of Life Sciences and Technology, Institut Teknologi Bandung, Bandung 40132, Indonesia; 5Department of Food Science and Technology, Faculty of Agricultural Engineering and Technology, IPB University, Bogor 16002, Indonesia; 6Department of Biology, Faculty of Mathematics and Natural Sciences, IPB University, Bogor 16002, Indonesia; 7Department of Physics, Faculty of Mathematics and Natural Sciences, Institut Teknologi Bandung, Bandung 40132, Indonesia; 8Department of Chemistry and Biomolecular Science, Faculty of Engineering and Graduate School of Engineering, Gifu University, Gifu 501-1193, Japan

**Keywords:** plant-derived exosome-like nanoparticles (PDENs), biomedical application, regenerative therapy, molecular cargo, anti-inflammatory molecules, biomolecules

## Abstract

Plant-derived exosome-like nanoparticles (PDENs) comprise various bioactive biomolecules. As an alternative cell-free therapeutic approach, they have the potential to deliver nano-bioactive compounds to the human body, and thus lead to various anti-inflammatory, antioxidant, and anti-tumor benefits. Moreover, it is known that Indonesia is one of the herbal centers of the world, with an abundance of unexplored sources of PDENs. This encouraged further research in biomedical science to develop natural richness in plants as a source for human welfare. This study aims to verify the potential of PDENs for biomedical purposes, especially for regenerative therapy applications, by collecting and analyzing data from the latest relevant research and developments.

## 1. Introduction

From a biomedical science perspective, a human tissue is a cellular hybrid system between cells and the complete organ [1]. Human tissue is an orchestrated organization composed of cells and their extracellular matrix (ECM) with specific composition and architecture, which carries out a specific function. Functional grouping of multiple tissues forms organs, in which a human body system is organized from an atomic scale, to molecular, macromolecular, organelle, cell, tissue, organ, and up to organ system scales. In the case of an injured, damaged, or missing tissue in the body, constructive remodeling and regenerative therapy are needed to provide engineered ECM with specific structural, mechanical, physical, and chemical properties that closely approximate to the replaced native tissue, which enables the remaining cells to attach, migrate, proliferate, differentiate, and regenerate for structural and functional replacement of the targeted tissue or organ [2,3,4]. This is the fundamental principle of tissue engineering.

In the framework of tissue engineering, the use of stem cells to increase tissue regenerative capacity has attracted scientists, especially mesenchymal stem cells (MSCs), due to the fact that their defining properties make them an ideal candidate to cure diseases. Although embryonic and fetal stem cells have the greatest potential to differentiate into different cell types, their application is limited due to ethical and safety issues [5,6,7,8,9,10], as well as the danger of unlimited and uncontrolled cells division [7,8,10,11,12,13,14,15]. Furthermore, the inherent heterogenicity and variation associated with cell expansion has become a major MSC limit for its clinical applications [16,17,18,19,20]. In addition, during in vitro cell processing and expansion, changes may occur to MSCs, thereby increasing the risk of MSC therapeutic application. Moreover, the risk of unwanted differentiation in vivo is a problem due to its clinical applications. This leads to alternative cell-free-based therapy with biomolecules secreted from MSCs, which are known as MSC-derived exosomes.

Extensive research has been carried out to identify the molecules involved in paracrine action of stem cells (SCs) for the opening of new therapeutic options in the concept of cell-free-based therapy. At this point, exosomes are nano-sized vesicular particles commonly secreted from eukaryotic cells into the extracellular space, and are intensively investigated as candidate therapeutic agents. These exosomes have known functions in cellular communication [19,20,21,22], nutrients [21,22], bioactive compounds delivery [19,20,22], and cellular immunity [19,20,22]. However, currently, there is no standardized procedure for the isolation, storage, and manufacturing of technology using a quality system for the safety of both donors and recipients in large-scale valorization of MSC-derived exosomes [19]. In the case of manufacturing, for instance, although extensive research has been conducted, the practical use of exosomes is still restricted by the limited exosome secretion capability of cells.

On the other hand, it is believed that plant-derived exosome-like particles (PDENs) have the potential to deliver nano-bioactive compounds to the human body [22]. As Indonesia is one of the herbal centers of the world, we are encouraged to develop natural richness for human welfare. In view of the current advancements and challenges, in this study, a comprehensive and concise review on the role of PDENs as a functional and beneficial biomolecule for biomedical applications and regenerative therapy is elaborated. To understand the potential of PDENs for biomedical applications and regenerative therapy, an overview of the origin, functions, and potentials of PDENs are herein described. Moreover, the important aspect on the strategy of combining PDENs and membranous scaffold containing ceramics is discussed in this study.

## 2. Plant-Derived Exosome-like Nanoparticle as Biomolecules

Exosomes were discovered in 1983 by Eberhard Trams and Rose Johnstone [23,24]. Exosomes are biologically small, nano-sized extracellular vesicles (EVs) with a diameter of 30–150 nm and density of 1.13–1.19 g/mL, secreted naturally by almost all eukaryotic cells [25,26,27,28,29,30,31]. Exosomes are important since these vesicles are similar to cargo-containing molecular components, such as DNA, RNA, lipid, and proteins, which are later released into the extracellular matrix as a form of intercellular communication [32,33,34]. Several studies have shown that exosomes comprise cytokines, transcription factor receptors, and other bioactive compounds [35,36].

The exact mechanism of exosome biogenesis remains controversial; however, they are generally synthesized in the multivesicular endosome compartments of the cell and released when these compartments fuse with the plasma membrane. Therefore, exosomes are also known as membrane-bound or membrane-derived vesicles [26,37,38,39]. Formerly, these exosomes were referred to as “ectosomes”, “shed vesicles”, and “small extracellular vesicles (sEVs)”. To date, however, the term “exosome” has received the most popularity [40,41,42]. In comparison to micro vesicles, apoptotic bodies, and oncosomes, exosomes are the smallest type of EVs [32,43,44,45].

Research proved EVs, such as exosomes, comprise their own specific cargo. This cargo becomes crucial since it represents the health and/or disease status of the source or parent cells, which has the capacity to alter recipient cells, both neighboring and distantly located ones, by adhering to the cell membrane and releasing its internal cargo. The release of this internal load induces physiological, phenotypic, and functional changes in the recipient cells [26,46,47,48,49,50]. These findings boost numerous investigations into the role of exosomes as novel biomarkers and potential therapeutic agents, which demonstrate that exosomes, as mediators of cell-to-cell communication, are related to physiological and pathological processes on the smallest scale, particularly between cells that occur in the living system [51,52].

Exosomes are divided into two categories: Natural and engineered exosomes. Natural exosomes are animal derived, mostly from mammals, and originate from cells [53,54,55,56] or biofluids [57,58,59,60], and plants as natural resources. Figure 1 and Figure 2 describe the biogenesis, sources, and contents of PDEN- and MSC-derived exosomes, whereas engineered exosomes are natural exosomes that have been artificially modified or loaded with therapeutic agents [39]. Animal-derived exosomes are further split into normal and tumor exosomes since exosomes can be produced in both normal and tumor conditions [39]. Studies related to mammalian-derived EVs, including exosomes, are documented in databases, such as Vesiclepedia (http://www.microvesicles.org, accessed on 17 February 2023) and ExoCarta at http://www.exocarta.org, (accessed on 17 February 2023) [61,62,63]. In addition to animals, plants are a source of exosomes, with good development prospects in the future. In recent years, studies have found that nano-sized EVs from plant cells have similar structures to the mammalian exosome [64,65,66,67], which is why these EVs are termed as “plant-derived exosome-like nano-particles” with various abbreviations, such as PDELNs [68], PLENs [69], or PDENs [70]. Other descriptive terms include “plant-derived nanovesicles (PDNVs)” [71], “plant-derived extracellular vesicles (PDEVs)” [72], “plant-derived exosomes” [73], “plant-derived edible nanoparticles” [74], or “edible plant-derived nanovesicles” [75]. This review uses the term “plant-derived exosome nanoparticles”, which is abbreviated as PDENs.

Plant-derived exosome-like nanoparticles are still not as popular as mammalian-derived exosomes, which have been extensively investigated, but the research trend has been increasing [71,75,76,77]. This positive trend is driven by the idea that consumption of certain foods or their associated components is often linked to health benefits and disease risk reduction [75]. Furthermore, concerns have been raised regarding the utilization of human exosomes for therapeutic chemical delivery since evidence stated that human exosomes may constitute one of our body’s scavenging processes, restricting large-scale production, and often carrying potentially transferable harmful substances, such as tumor-derived molecules and foreign nucleic acid [78,79,80,81,82,83,84]. Similar to other allograft materials, the large-scale manufacturing and usage of exosomes derived from humans will face obstacles in the future due to exosome sources and ethical concerns.

In addition to plant-derived micronutrients, sterols, fibers, and phytochemicals, studies on plant-derived bioactive compounds and their impact on health have recently been broadened to include EVs [75]. The Food and Agriculture Organization (FAO) division of International Network of Food Data Systems (INFOODS) collects data on foods containing exosomes (or endosomal-derived vesicles) in their databases; https://www.fao.org/infoods (accessed on 17 February 2023). The database contains the following information regarding the plant-derived exosomes: Common name, group, scientific name, sample type, isolation method, and references [72,85]. PDENs, similar to mammalian-derived exosomes, comprise a variety of components, such as proteins, lipids, mRNA, microRNA, and their unique source-dependent bioactive constituents [86,87,88]. PDENs are composed of cytosolic and membrane proteins. They contain numerous types of confirmed proteins, such as heat shock proteins, which are accepted as exosome markers (e.g., CD9 and CD63), transmembrane proteins (e.g., actin, proteolysis, aquaporin, and chloride channels proteins), defense proteins, and other plasmalemma-associated proteins [89,90,91,92,93]. Research substantiated that PDENs from citrus fruits comprise proteins involved in diverse activities, including fructose bisphosphate for glycolysis, HSP80 and PTL39 for protein folding and transport, and PTL3 and clathrin-3 for cell growth and division. In addition, enzymes and antioxidants were identified [94]. In terms of protein level, PDENs are lower than mammalian exosomes [89].

Lipids in EVs can induce cellular responses in recipient cells, maintaining the structural stability of exosomes, facilitating their cargo components (uptake and retention), and playing an important role in intercellular communication [95,96]. PDENs are rich in phospholipids (e.g., phosphatidic acids (PA), phosphatidylethanolamines (PE), including plant lipids, such as galactolipids (e.g., monogalactosyldiacylglycerol (MGDG) and di-galactosyl di-acylglycerol (DGDG)) [89,97]. Based on previous studies, lipid composition approximations of some edible plants have been reported, such as exosome-like nanoparticles derived from ginger, which comprised 40% PA, 30–40% DGDG, and 20% MGDG [89,95], turmeric comprised 42% DGDG, 12% MGDG, 20% PA, and phosphatidylcholine (PC) as reported by Liu et al. (2022) [98], orange juice comprised 40% PE, 25% PC, and 5% PA [99], and grapefruit comprised 45.52% PE and 28.53% PC [65]. From the findings, it was known that the amounts of lipid compositions in PDENs vary depending on their sources. Moreover, it was observed that PDENs are different from mammalian exosomes since they are phospholipid-rich but cholesterol-free, whereas animal exosomes are cholesterol-rich [100,101].

Plant-derived exosome-like nanoparticles often comprise RNAs and a significant quantity of microRNAs (miRNAs), a class of small and noncoding RNAs (17–24 nucleotides). It has been found that miRNAs are capable of disrupting mRNA translation and transcription [102,103,104]. Moreover, based on previous studies regarding the potential for cross-kingdom communication by PDENs through gene expression modulation [105], which involves miRNA, it has been found that the internalization of PELNs is possible in plant cells, mammalian cells, fungi, and bacteria [101,106,107,108]. In addition, several bioactive compounds are found in PDENs, which have the potential to determine their effect. This naturally occurring metabolite varies not only based on plant origin, but also on the EV population size, preparation form, and isolation technique. Table 1 shows the compounds found in PDENs from several sources. In addition, PDENs comprise myo-inositol, quinic acid, and aucubin, which are anti-cancer and anti-inflammatory compounds [109].

PDENs comprise 6-gingerol, 8-gingerol, 10-gingerol, and 6-shogaol, which are found in ginger [89,113]; naringenin found in grapefruit [65]; citrate, vitamin C, and galacturonic acid-enriched pectin-type polysaccharide found in lemon [66,110]; vitamin C found in strawberry [88]; epigallocatechin gallate, epicatechin gallate, epicatechin, vitexin, myricetin-3-*O*-rhamnoside, kaempferol-3-*O*-galactoside, and myricetin found in tea flower [66,111]; fiber β-glucan found in oat; and sulforaphane found in broccoli [112]. In addition, PDENs comprise myo-inositol, quinic acid, and aucubin, which are anti-cancer and anti-inflammatory compounds [109]. 

Plant-derived exosome-like nanoparticles need to undergo a series of processes before they can be used. After plant extraction, the first step is isolation and there are several isolation techniques used for PDENs. Most isolation methods are based on density, surface content, size, or precipitation [114]. Based on density, frequently used techniques are differential ultracentrifugation and density gradient centrifugation. These techniques are simple and cost-effective; therefore, they are considered as the “gold standard” for the isolation method [74,114,115,116]. Differential ultracentrifugation is the consecutive removal of particles based on size and density using progressive centrifugal durations and forces. The procedure is carried out via a step-by-step methodology, starting from the separation of cells, and then gradually cell debris, apoptotic cells, and micro-vesicles. As the remaining particles become smaller, the spinning speed is gradually increased (100,000–200,000 g) throughout the process [117,118,119]. The formation of sediment vesicles, protein aggregates [4], and exosomal aggregates [74,120], the extended duration, labor intensiveness, and the necessity for highly expensive equipment are the major drawbacks of this procedure [121,122,123,124]. In contrast to differential ultracentrifugation, density gradient ultracentrifugation isolate PDENs are based on size, mass, and density. This method differentiates between PDENs and contaminants with different densities using a subsequent phase with a sucrose density gradient (10–90%) or other material (e.g., iodixanol) [125,126]. Furthermore, density gradient ultracentrifugation is still extensively used today since this method increases the final purity of PDENs isolate [74,117]. Nevertheless, this technique yields few exosomes, is time-consuming, and requires expensive equipment and specialized knowledge [127,128].

Sized-based isolation methods including ultrafiltration and size exclusion chromatography (SEC) are commonly used to separate PDENs. Ultrafiltration, a faster alternative to ultracentrifugation, comprises the use of membrane filters with different molecular weight cutoffs (MWCO) and pressure to remove large molecular size impurities, followed by exosome isolation [67,129]. In this method, larger sample components including cells, cell debris, and macromolecules are not permitted to pass through the membrane pores [39]. Utilizing filters with pore sizes of 0.80 and 0.45 microns, larger particles can be eliminated. After collecting the remaining sample, a filter with pores of smaller diameters than exosomes (0.22–0.1 m) is used to remove particles smaller than exosomes. This approach obtains exosomes between filters with maximum and minimum pore diameters [130]. Ultrafiltration is simple and efficient without affecting the exosomes activity technique; however, it is low in purity and not suitable for wide usage [131]. SEC is another size-based method using gel-filtration. The mobile phase is an aqueous solution that transports the sample along the column, while the stationary phase is a porous filtration polymer that permits differential elution. With the support of gravity, larger particles are eluted before smaller ones. However, other factors, such as molecular weight and shape, can also influence this separation process [21,115,132]. The advantages of this technique include good homogeneity and preservation of the integrity, morphology, and functionality of nanoparticles. However, this technique is not widely used due to low purity production and absence of selectivity in extracting exosomes from particles of almost identical size [133,134,135,136]. Asymmetric flow field-flow fractionation (AF4) is a recent, size-based isolation method. AF4 separates nanoparticles with sizes ranging from a few nanometers to an undetermined amount of micrometers, depending on their hydrodynamic diameters, in a thin, flat channel with a semipermeable wall membrane [136].

Moreover, the polymer precipitation method can be used to isolate PDENs, although this method is originally used to isolate viruses [137,138,139]. The scientific reasoning is due to the fact that both viruses and small PDENs have similar biophysical characteristics [39]. This method uses polyethylene glycol (PEG), specifically PEG6000, which is capable of forming a net-like structure to trap PDENs before precipitation. These precipitated and aggregated PDENs are harvested through lower centrifugation rates [140]. This technique is easy to operate, time-efficient, and capable of processing large doses of samples [39]. Furthermore, this study illustrates that this technique is less expensive and outperforms ultracentrifugation in terms of purity and recovery [141]. Polymer precipitation using concentration-adjusted PEG6000 produced PDENs with a smaller particle size than ultracentrifugation [142]. However, impurity and low recovery time are the primary disadvantages of this technique [39,143].

Another method for isolating PDENs is the affinity-based isolation method or immunoaffinity chromatography (IAC). IAC is a separation and purification method based on the specific interaction of antibodies and ligands to isolate desired components from the heterogenous mixture [39,115]. The biomarker or antigen should be a high-abundance protein on the surface of exosome membranes, such as the four-transmembrane protein superfamily and ESCRT complex-related proteins [39,143]. Several methods have been proven to be successful for isolation-related IAC, among these are magnetic beads [144] or FeO_3_ nano-cubes [145] loaded or coated with antibodies, as well as heat shock proteins [146], heparin [147], or epithelial cell adhesion molecules [148]. These methods accommodate high-purity exosomes and selectively isolate exosomes that carry a specific marker [114]. Nevertheless, the drawbacks of these methods include the heterogenous nature of PDENs and the loss of exosomes integrity by removal of the antibody process [149,150]. In addition to the aforementioned common methods, few other advanced methods can be used to isolate the PDENs, such as microfluidics-based isolation by trapping PDENs in a porous microstructure with a selective micropillar design of 40–100 nm in diameter [151]. This can be performed via the acoustics-based approach using a combination of microfluidic chip and acoustic waves [152], or a commercially available exosomes isolation kit that is time-saving but costly, such as total Exosome Isolation kit (Invitrogen), exoEasy Maxi kit (QIAGEN), MagCapture™ Exosome Isolation Kit PS (Wako), Eloquence (System Biosciences), Exo-spin (Cell guidance systems), and Minute™ Hi-Efficiency Exosome Precipitation Reagent [39,153]. These kits can be used to separate exosomes from biological samples, such as serum, plasma, and CSF. However, the purity, amount, and size distribution of the collected exosomes are very different [154,155].

Plant-derived exosome-like nanoparticles, such as mammalian exosomes, can be characterized using the ultrasensitive microscopic technique, such as scanning electron microscopy (SEM), transmission electron microscopy (TEM), atomic force microscopy (AFM), or using cryo-electron microscopy for structural analysis in subcellular level [119,156]. Optical characterization approaches cannot identify PDENs due to their small diameters; therefore, various methods of electron microscopy are essential to determine their morphological size and shape [157]. SEM can be used to observe the surface structure; however, it is lower in resolution. Meanwhile, the high-resolution TEM provides information regarding the internal structure and particles distribution; however, it is not suitable for rapid measurement since the pre-processing step is more complex [158,159]. Sample preparation for TEM proved to be responsible for the observed morphology variations [160,161].

Particles charge and size of PDENs can be observed using dynamic light scattering (DLS); however, this technique cannot detect the concentration and is not suitable for measuring complex exosomes with large size range. The concentration of PDENs can be measured using the nanoparticle tracking analysis (NTA) [39,119]. This technology emerged as the “gold standard” for exosomes characterization since PDENs can be observed in real-time due to their fast detection speed; however, the operation is complex and can affect the PDENs quantification [39,135,162]. To determine the size and concentration of PDENs in suspension, resistive pulse sensing (RPS) technology can be used. Since a voltage is applied, the detection of exosomes by RPS with the transfer of single nanoparticles through the nanopore is established [156,163]. Western blot and enzyme-linked immunosorbent analysis (ELISA) can be used to detect the expression of PDENs marker proteins [41,164]. Flow cytometry can be used to detect the biomarker of PDENs. Flow cytometry is a high-throughput technology and capable of high-speed multi-channel analysis with low sample concentration [39,156]. However, the newest generations of digital flow cytometry instruments, with a lower limit of measurement of 100 nm, are still unreliable in terms of accuracy and resolution [156]. Another technique to characterize PDENs specifically for the determination of the PDENs weight can be performed using the developed nanomechanical resonator. Moreover, zeta analyzer can be used to observe the repulsive nature against aggregation or dispersity and the membrane potential [119,165]. 

Similar to the most common exosomes, PDENs cannot be maintained for a long period of time. Therefore, it is necessary to preserve PDENs in order to maintain their biological activities and for readily accessible clinical application. Plant extracts stored prior to PDENs isolation may influence the separation, content, and function of PDENs, such as those reported in EVs derived from biofluids. Before the isolation process, samples were mainly stored in freezing conditions for short or long periods. Although 4 and −80 °C or lower are mentioned and generally used in laboratories today, the problem is that no optimal storage condition has been determined for isolated EVs [166,167,168,169,170,171]. EVs may alter to varying degrees during storage, resulting in size, shape, function, and content loss [172]. Current techniques to accommodate this include cryopreservation, freeze-drying, and spray-drying [39,173].

Cryopreservation is the application of low temperatures below those needed for biochemical reactions (4 °C, −80 °C, or −196 °C) to preserve the functional stability of biological particles, including PDENs, using cryoprotectants [166,172,173,174]. There are two types of cryoprotectants which are differentiated based on their permeability or penetration and both aimed to reduce the injury of EVs. Penetrating cryoprotectants (e.g., glycerol, DMSO, and ethylene glycol) have a small molecular weight that can penetrate the cell membrane, while non-penetrating cryoprotectants (e.g., sucrose, mannose, and trehalose) form a hydrogen bond with water to prevent ice crystal formation [159,175,176]. A previous study has proven that cryopreservation in −80 °C with or without cryoprotectants, such as trehalose 25 mM, DMSO 6 and 10%, and glycerol 30%, can maintain the concentration of EVs up to 6 months [176]. However, storage conditions may vary depending on the source of EVs. Regarding safety, trehalose as a non-penetrating type, is the best option and most effective antifreeze [175,177].

Freeze-drying process is a technology that involves sublimation and desorption, where moisture-containing materials are frozen to a solid below the freezing point prior to their use [39,172]. Freeze-drying is a new technique for preserving EVs, including PDENs, and the optimal storage temperature for freeze-dried EVs is 4 °C [178,179]. Freeze-drying is advantageous in the expansion of the lifespan of heat-sensitive materials, such as EVs, vaccines, and proteins since it can maintain the material’s original activity, while causing less damage to biological tissues and cell bodies. Moreover, the material can be easily stored in a constant state and reconstituted by simply adding water. However, due to the freezing and dehydration pressures generated during the freeze-drying process, the molecular structure of the biomolecule may be destroyed [39,172]. Based on previous studies, cryoprotectant products, such as trehalose, may be used to prevent aggregation during the freeze-drying process and reduce the alteration in the stability and morphology of EVs, including PDENs [179,180,181]. Spray-drying, on the other hand, is an option. Spray-drying is a one-step method that is simpler than freeze-drying [172]. This technique is more economical, can be used with a variety of agents, and allows for product size adjustment. First, PDENs solution is atomized in a drying room, and then when moisture interacts with hot air, it quickly vaporizes, resulting in dry powders [39,174]. PDENs stability can be affected by factors, such as EV solution feed, atomization pressure, and output temperature. Furthermore, residual moisture may cause chemical instability by lowering the solid particle state’s glass transition temperature [172].

The advantages of PDENs as biomolecules are: (1) Provided by nature and can be involved in intercellular communications, (2) phospholipid-rich characteristics in order that the lipid membrane can protect the cargo from the external agent, which can deteriorate the bioactive compound inside the PDENs, (3) natural mechanism of cellular uptake in order that they may have the ability to pass through some of the barriers in our body, such as blood brain barrier and the placenta, (4) tolerated by the immune system as they are currently present in foods ingested by humans, (5) verified scalability and suitability for industrial applications, (6) non-toxic, as they are produced from organic fruits and vegetables [71], and (7) long-term availability from numerous types of plants, which can be cultivated. These PDENs are worth development in various fields, especially in the field of biomolecular medicine. This can be accomplished by exploring the cargo, in particular, the contents and functions of bioactive constituents. Then, they can be related to the necessary various problem-solving strategies. A better understanding of PDENs in the future has the potential to usher in a new paradigm for natural medicine, which employs compounds that are abundantly available, more effective, efficient, and have significantly fewer adverse effects than the currently accessible medications.

Exosomes have been investigated in animal models and clinical trials in recent years to obtain their biological properties. Grape-derived PDENs have been proven to be nontoxic to intestinal macrophages, splenic, and liver cells of mouse model [65,100]. Ginger-derived PDENs are nontoxic to colon-26 epithelial-like cell lines, and RAW 264.7 macrophage-like cell lines in the colon’s mouse model [89]. Another study proved that citrus lemon-derived PDENs inhibit the growth of CML tumors in vivo by reaching the tumor site and activating TRAIL-mediated apoptotic cell processes in the liver, spleen, and kidney of mouse model [182]. Moreover, ginseng-derived PDENs proved to be nontoxic to BMDMs (bone marrow-derived macrophages), B16F10 (mouse melanoma cell line), 4T1 (mouse mammary carcinoma line), and HEK293T (human embryonic kidney cell line) in the liver and spleen of mouse model [81]. The development of human exosomes and PDENs as potential biomolecules has reached the clinical trial stage [183,184]. According to a ClinicalTrials.gov survey (https://clinicaltrials.gov/, accessed on 17 February 2023), exosomes are used for clinical trial as biomarkers (50%), exosome-therapy (28.44%), drug delivery systems (5.17%), cancer vaccines (1.72%), and other analysis (14.66%) [183]. Two reported clinical trials using PDENs aimed to encapsulate curcumin into grape-derived (NCT01294072) PDENs and ginger-derived (NCT04879810) PDENs for the treatment of colon cancer and irritable bowel disease, respectively. Another clinical trial using grape-derived PDENs (NCT01668849) aimed to investigate the ability of grape-derived PDENs to prevent oral mucositis associated with head and neck cancer chemoradiation treatment. However, none of these trials have been completed, and the results are not yet available [183,184]. Furthermore, although several experimental studies have been documented, there is less evidence on GMP manufacturing for plant-derived exosomes [21,185].

## 3. Biomedical Applications of PDENs

The literature search strategy using PubMed database was administered to find research related to the use of PDENs for biomedical applications. The search was conducted on PubMed database for published articles covering “plant-derived exosome-like nanoparticles” in their title and/or abstract, published up to 7 December 2022. The search method uncovered 42 literatures, of which 21 were excluded due to their irrelevance to biomedical applications (7 articles) or were written in a form of a review article (14 articles). The 21 research articles reported various findings on numerous PDENs sources, as summarized in Table 2 for further analysis on the recent status of PDENs utilization in biomedicine. 

As shown in Table 2, 21 studies on the application of PDENs in biomedical fields reported a distinct capacity of the vesicles to act as a potential biomedical therapeutical agent. From the various studies conducted, PDEN-derived contents were reported to interact with host cells to regulate and modulate cellular activities. Generally, most of the studies reported reduced various inflammatory mediators, oxidative stress markers, and tumor growth, whether in vitro or in vivo. There are also studies observing PDENs uptake by bacterial cells in *Porphyromonas gingivalis* [186], *L. monocytogenes*, and *L. monocytogenes*-EGD [187]. 

**Table 2 biomedicines-11-01053-t002:** Summary of research articles on PDENs in biomedical approaches.

PDENs Source	Type of Study	Findings	Refs.
*Phellinus linteus*	In vitro, in vivo, and clinical trial	PDENs stimulate anti-aging effects through the promotion of COL1A2 and inhibition of MMP-1, ROS, MDA, and SA-β-Gal in UV-induced aging skin cells	[188]
Coffee	In vitro	PDENs promote proliferation of LX2 and HEP40 cell lines in a concentration-dependent manner exerted by their miRNA contents	[189]
*Momordica charantia*	In vivo and in vivo	PDENs inhibit MMP-9 and apoptosis, and activate AKT and GSK3β phosphorylation, thus preserving the blood brain barrier to act as a neuroprotective agent	[190]
*Beta vulgaris*	In vitro	PDENs show antioxidant activities, angiogenesis, and inhibition of fibroblast migration to prevent scar formation and exert anti-aging effects	[191]
Rice	In vitro and in vivo	PDENs and their miRNA contents enhanced cell proliferation and GLUT1 expression to regulate blood glucose and metabolism	[192]
Hawaiian ginger roots	In vitro and in vivo	PDENs inhibit NF-κB-mediated inflammation and apoptosis, and limit SARS-CoV-2 S and Nsp12 expression through their miRNA contents	[185]
Blueberry	In vitro and in vivo	PDENs inhibit oxidative stress by attenuating ROS, Bcl-2, and HO-1, and accelerating Nrf2 translocation, thus ameliorating insulin resistance and liver dysfunction	[193]
*Asparagus cochinchinensis*	In vitro and in vivo	PDENs are internalized through phagocytosis and inhibition of cancer cells growth. PEGylated PDENs show promoted blood retention and anti-tumor capacity than bare PDENs	[194]
Strawberry	In vitro	PDENs accumulate in the cytoplasm, promote cell viability, and exert antioxidant activity through their vitamin C constituents	[88]
Blueberry	In vitro	PDENs inhibit TNF-α-induced cytotoxicity and oxidative stress, as well as mRNA expression of IL1R1, IL-6, MAPK1, ICAM1, TLR8, TNF, HMOX1, and NFR1 post-TNF-α administration	[195]
Shiitake mushroom	In vitro and in vivo	PDENs attenuate NLRP3 activation through the inhibited Casp1 p10 level, and inhibit IL-1β protein assembly through the significant reduction in pro-IL-1β, leading to liver protection	[196]
Ginger	In vitro and in vivo	PDENs are internalized by P. gingivalis and inhibition of bacterial growth through interaction with HBP35 and membrane depolarization thus inhibiting, bacterial attachment to oral epithelial cells and periodontal bone loss	[186]
Ginger	In vitro, in vivo, and clinical trial	PDENs internalization by gut microbiome regulated by its lipid composition, whereas its RNAs affect bacterial metabolisms and alter pro-inflammatory cytokines profile	[185]
Blueberry, coconut, ginger, grapefruit, Hami melon, kiwifruit, orange, pea, pear, soybean, and tomato	In vitro	PDENs comprised of miRNAs that could directly interact with mammalian mRNAs of inflammatory mediators	[197]
*Citrus limon*	In vitro and in vivo	PDENs inhibited tumor cell viability, but provided no inhibitory effects on normal cells. PDEN promotes Bad and Bax genes, and reduces Survivin and Bcl-xL in tumor cells, thus inducing TRAIL-induced cell death and dampening tumor growth	[182]
*Aucklandia lapp, Rhodiola crenulata* and *Taraxacum mongolicum*	In vitro and in vivo	PDENs have anti-fibrotic and anti-inflammatory effects through their sRNA, thus ameliorating lung fibrosis and inflammation	[198]
Ginger roots	In vitro	PDENs modification with arrowtail RNA was able to deliver and target specific cells with significant reduction in survivin expression and tumor growth	[199]
Grapes, grapefruit, ginger, and carrot	In vitro and in vivo	PDENs internalized by macrophages and stem cells promote HO-1, IL-6, and IL-10 expression, Nrf2 translocation, and activate TCF4 transcription through Wnt activation in the gut tissue	[200]
*Citrus sinensis*	In vitro	PDENs protect their miRNA contents from human salivary degrading enzymes to act as a delivery agent	[201]
Ginger	In vitro	PDENs significantly reduced TNF-α, IL-8, and IL-1β mRNA level by inhibiting NF-κB in intestinal cells through their regulatory miRNAs	[202]
Mulberry bark	In vitro and in vivo	PDENs enhanced AhR-mediated pathway mediated by HSP70, leading to COPS8 induction and inhibition of bacterial mRNAs in gut tissue, thus reducing the level of IL-6 and IL-1β	[187]

Abbreviations: PDENs = plant-derived exosome-like nanoparticles; MMP1 = matrix metallopeptidase 1; ROS = reactive oxygen species; MDA = malondialdehyde; SA-β-Gal = senescence-associated β-galactosidase; UV = ultraviolet; miRNA = micro RNA; MMP-9 = matrix metallopeptidase 9; GSK3β = glycogen synthase kinase-3 beta; GLUT1 = glucose transporter 1; NF-κB = nuclear factor-κB; Nsp12 = nonstructural protein 12; Bcl-2 = B-cell lymphoma 2; HO-1 = heme oxygenase-1; Nrf2 = nuclear factor erythroid 2–related factor 2; PEG = polyethylene glycol; TNF-α = tumor necrosis factor alpha; IL1R1 = interleukin 1 receptor type 1; IL-6 = interleukin 6; MAPK1 = mitogen-activated protein kinase 1; ICAM1 = intercellular adhesion molecule 1; TLR8 = toll-like receptor 8; HMOX1 = heme oxygenase 1; NFR1 = nuclear respiratory factor 1; NLRP3 = NLR family pyrin domain containing 3; Casp1 = caspase 1; IL-1β = interleukin 1 beta; HBP35 = hemin-binding protein 35; RNA = ribonucleic acid; Bad = BCL2 associated agonist of cell death; Bax = Bcl-2 associated X-protein; Bcl-xL = B-cell lymphoma-extra-large; TRAIL = tumor necrosis factor-related apoptosis-inducing ligand; sRNA = small RNA; IL-10 = interleukin 10; TCF4 = transcription factor 4; IL-8 = interleukin 9; AhR = aryl hydrocarbon receptor; HSP70 = heat shock protein 70; COPS8 = COP9 signalosome complex subunit 8.

As described in Table 2, ten studies reported immunomodulatory activities of PDENs derived from various sources. From these studies, it can be inferred that PDENs control inflammation from the transcriptional levels of mRNA, as some of these studies reported a lowered level of mRNA in various inflammatory markers. PDENs inhibit various intracellular signaling, such as AKT, GSK3β [190], TCF4, Nrf2 [200], NF-κB [185], and AhR [187], leading to altered transcriptional signaling and modulated downstream protein expressions. De Robertis et al. (2020) and Yin et al. (2022) reported hindered mRNA level of TNF-α, IL-1β, IL1R1, IL-6, IL-8, MAPK1, ICAM1, TLR8, HMOX1, and NFR1 after PDENs administration [195,202]. This leads to the lowered quantity of cytokines secretion in ELISA assays [187]. Transcriptional regulatory activities of PDENs might be conferred from PDENs constituents, such as lipids, proteins, and especially miRNAs. It was reported that plant-derived miRNAs could interact with RNAs in human cells to regulate cellular behavior [197]. Teng et al. (2021) reported that PDEN-derived miRNAs could inhibit activation of NF-κB, leading to an attenuated cytokine profile [185]. In the case of inducing transcription, PDEN-derived miRNAs can bind the UTR’3 of miRNA to repress translation and bind the promoter region to induce transcription [103]. This leads to the potential of PDEN-derived miRNAs to be utilized as endogenous drugs for the alleviation of disease symptoms or prevention of disease onset from transcriptional and translational levels.

The antioxidant is among the other biological activities of PDENs, as reported in five studies (Table 2). Antioxidant properties of PDENs might be attributed to their ability to deliver miRNAs to cells. Han et al. (2022) reported that miR-CM1 in mushroom-derived PDENs could target the 3′UTR of Mical2, and thus inhibit their expression to reduce ROS. Mical2 deactivation would lead to the inhibition of cellular injury and damage caused by oxidative stress, and thus exert anti-aging effects on skin cells [188]. This statement was corroborated with another study, which reported that the DPPH radical scavenging assays and FRAP assays of PDENs are comparable to ascorbic acid; however, it could only exert around half of the potential of Beta vulgaris pure extract. Despite the fact that Beta vulgaris PDENs contain only half of the amount of the antioxidant compared to the extract, PDENs were reported to provide no significant differences with the extract in terms of tumor cells toxicity and normal cells viability. This would provide the percentage of wound closure, which is significantly higher than the pure extract [191]. PDENs were also found to enhance the mitochondrial content to further inhibit MMP-1 production and apoptosis-associated proteins, promote anti-oxidant enzymes transcription through Nrf2, and protect cells from inflammatory damage through HMOX1 modulation [193]. Regulation of oxidative stress by PDENs would further inhibit apoptosis and enhance cellular proliferative activities. These activities might not only be due to their miRNA contents, but also the other constituents found in PDEN-derived biomolecules, such as vitamin C [87]. 

PDENs have also been reported to control tumor growth in three studies. MTT assays were carried out on various cancerous cell lines after exposure to PDENs, and have shown growth inhibition on the cells with varying IC50 values for each cell [182,194]. Then, flow cytometry analysis has shown that the growth inhibition would be further followed with apoptosis of the cells until a ratio of 46.5% for PDENs concentration of 40 μg/mL [194]. These effects might result from various pathways, as previously reported. Internalized PDENs could enhance Bad and Bax, which are pro-apoptotic genes, and reduce survivin and Bcl-xL, which are anti-apoptotic genes in cancer cells [182]. Zhang et al. (2021) reported the same results in different sources of PDENs. Activation of Bad and Bax were reported to further promote Caspase 9 to induce cancer cells apoptosis through cellular proteins cleavage. Moreover, PDENs have been reported to induce cancer cell death by upregulating TRAIL and its receptor, Dr5 [194]. These cellular effects of PDENs might be disadvantageous if the PDENs are inhibited by normal cells. Therefore, surface modification of PDENs has been developed to deliver PDEN-derived nucleic acids to specific target cells. Li et al. (2018) have successfully built PDENs construction with folic acid on the surface to facilitate PDENs cellular uptake, leading to substantial genetic alteration of cancer cells and tumor growth inhibition [199]. This marks the potential of PDENs to be engineered to suit various biomedical therapies with specific requirements and conditions. Furthermore, these findings have been corroborated by the study of Boccia et al. (2022), which mentioned the anti-tumor activity of extracellular vesicles from *Salvia dominica*’s hairy roots through accumulation in the cytoplasm of cancer cells and inducing cancer cell death reprogramming, leading to cytotoxicity [203].

Based on the literature searches and analyses carried out in the biomedical database, it can be assumed that PDENs are emerging biomolecules with the most promising therapeutical agents, especially as an anti-inflammatory agent and antioxidant. Despite the promising biological activity, it was noticeable that PDENs are not currently explored for their potential in the field of biomedicine. Therefore, exploration on the promising strategies of PDENs utilization in biomedicine needs to be conducted thoroughly.

Table 3 lists the cells used in the PDEN-related studies, which shows that PDENs research currently involves human cells, especially tumor cells, fibroblast cells, and epithelial cells. The results of the studies show that the potential of PDENs varies as anti-tumor [194], anti-aging [188,191], and anti-scaring agents [191]. Moreover, research on PDENs internalization by bacterial cells shows a promising scientific breakthrough. Although further investigations are needed, PDENs were found to control microbiota and inhibit dysbiosis to prevent phenotype changes in commensal bacteria [186,187]. Moreover, it can be inferred that PDENs often associate with applications on soft tissue cells, showing the research gap of their utilization in bone tissue cells. 

Excluded review articles were further analyzed to observe statements regarding undisclosed prospects and challenges of PDENs, which were analyzed to find the areas for further research to establish better strategies on PDENs utilization in the future. Review articles with no statements of prospects and challenges of PDENs applications were not listed in Table 4. 

As described in Table 4, the review articles explored various future approaches of PDENs fabrication for applications in disease therapies. Most of the conducted reviews concluded the potential of PDENs as an alternative prominent drug-delivery system due to their stability in body fluid and capability of enhancing cellular uptake with minimum cytotoxicity. The potential of PDENs uptake by bacterial cells would also contribute to microbiome modulation and regulation, in oral cavity or intestine, to prevent infection and enhance local immunity [69,72]. Abundance of PDENs in plants makes it a great prospect for functional food development in the future, as supported by their stability in digestive lavage [21].

## 4. Roles of PDENs in Regenerative Therapy

It has been reported that PDENs can alleviate inflammation and attenuate tissue destruction [185,188]. However, their potential for tissue regeneration and engineering, especially on hard tissue, was never fully discussed. In contrast to MSC-derived exosomes [208,209,210,211], research on the regenerative capacity of PDENs seems to not have been carried out comprehensively. Despite the lack of direct and complete evidence, PDENs seem to be able to exert their bioactivity in order to act as a potential regenerative therapeutic agent in the principle of tissue engineering. 

In regard to the tissue engineering paradigm, inflammation regulation has a very significant role toward tissue regeneration. Furthermore, immunocompetency is the key that affects the loss and success of regenerative processes [212]. This is due to the fact that low inflammation enables tissue destruction by pathogens, while high and chronic inflammation leads to pathologic conditions. Both conditions result in altered tissue regeneration. Therefore, balancing inflammatory processes is a promising approach to promote better tissue regeneration. From this point of view, immunomodulatory activities of PDENs have become a necessity to provide ideal microenvironments for tissues to regenerate [213]. 

Furthermore, it has been reported that plant-derived miRNAs comprise immunomodulatory capacities through TLR3 binding in dendritic cells and interferences to TRIF signaling, which limit T cell proliferation, leading to restricted pro-inflammatory cytokines production. Chronic inflammation, for example, often associates with fibrotic repairs that are caused by the large production of pro-inflammatory cytokines in acute wound; therefore, an accelerated resolution is needed for proper regeneration to take place [214]. These activities were reported to be not plant specific [215]. Furthermore, it was known that PDENs have other advantages than regular anti-inflammatory drugs, as they can activate pro-inflammatory mediators and clear the microenvironment. The dsRNA, ssRNA, and siRNA in PDENs could interact with pattern recognition receptors and activate immune cells [212,213,215].

Cho et al. (2022) reported promoted skin regeneration gene markers and wound healing after PDENs administration. Referring to the previous studies [216,217], PDEN-derived miRNAs could potentially perform cellular reprogramming to direct macrophage polarization, and thus enhance inflammation to reach the resolution. Another study also found that Aloe Saponaria-derived PDENs could repress pro-inflammatory mRNAs, enhance fibroblasts migration and proliferation, and further promote angiogenesis to regenerate skin tissue [218]. PDEN-derived miRNAs were also found to possibly regulate the plasticity of remaining viable cells at the injury site and modulate the extracellular matrix (ECM) to initiate cell differentiation, proliferation, and tissue regeneration [217]. Furthermore, it has been reported that PDEN-derived miRNAs could be obtained by bone marrow-derived mesenchymal stem cells. This process enhances neural differentiation markers signaling, and thus regenerates the nerve tissue to retrieve the sensory function [219]. 

Antioxidative capacity of PDENs also indirectly plays important roles in tissue regeneration [220]. The injured tissue and cell produce ROS that would lead to activations of various pro-inflammatory transcription factors, such as Jun N-terminal kinase. In addition, cells would activate Nrf2 transcription to pursue oxidative stress balance [213]. Conditions with overproduced and prolonged oxidative stress could damage cellular contents, leading to altered regeneration. At this point, antioxidative activities of PDENs have become a promising strategy to enable tissue regeneration and proper wound healing [221]. Antioxidative capacity of PDENs can also promote cell proliferation and wound healing through Nrf2 activation to maintain cellular redox homeostasis, as shown in scratch assays [222,223]. Moreover, the high number of miRNAs in PDENs might play roles in oxidative stress homeostasis, as miRNAs could modulate redox sensors, antioxidant activity, and oxidative stress regulations in cells [224]. Furthermore, some miRNAs were reported to be modulated by the antioxidants level. Consequently, homeostasis between miRNA, antioxidant, and oxidative stress could indirectly affect the regenerative capacity of the cells [225]. 

On the other hand, to exert their functions, PDENs are required to interact with residing cells. The interactions might be controlled and affected by particle size and surface charge. It was reported that ginger-derived exosomes isolation with PEG provides the smallest size of 278 nm, with the highest size frequency of 3795 nm, along with 1664 nm of average varieties of ginger and quinoa-derived exosomes. Whereas quinoa-derived exosomes have a smaller size average of 445 nm, with the lowest size of 82 nm. The particle size is observed to be affected by the handling and procedures used during PDENs isolation [22]. A considerably large heterogeneity in the particle size of PDENs analyzed by nanoparticle tracking analysis, dynamic light scattering, or electron microscope might be diverging among studies [218,226]. In general, the current report shows that PDENs have a significantly larger size than MSC-derived exosomes, in the range of 50–200 nm [227]. This large particle size was reported to reduce PDENs bioactivity by significantly hampering the cellular uptake rates, especially when compared to MSC-derived exosomes [22]. Despite the hindered internalization rate, the high amount of PDENs were still able to reach intracellular compartments of various human cells [218,228]. Size modification and engineering of PDENs might be one of the most challenging strategies to enhance their bioactivity and uptake rate.

## 5. Strategies for PDENs Application in Tissue Engineering

Exosomes are emerging sources for cell-free regenerative therapy for the past decades, as they provide better clinical benefits than cell-based therapy [19,209]. Abundant and high accessibility to sources of PDENs have large advantages in production manufactures and clinical practices. Engineered exosomes using biomaterials potentially contribute better clinical benefits compared to pure exosome therapy, since biomaterials could direct, regulate, and enhance specific cellular behavior for specific clinical demands [229]. Since PDEN-based tissue engineering has become a promising research interest to develop better alternatives in regenerative medicine, exploration on its potential integration with biomaterials is needed. 

Fine-tuned bioactive scaffolds with incorporated biomolecules are demanded as they could actively direct and regulate specific cellular interaction, preserve, or enhance bioactivity, enable long shelf-time, and maintain controlled-release [230]. However, PDENs delivery to the injury site would need an ideal and adequate delivery system, as unregulated release and unspecific cell-targeting could hinder therapeutical results. In view of this approach, biomaterials are often used to act as retention media for bioactive molecules, such as exosomes, while providing a stable and ideal microenvironment for resident progenitor cells to proliferate and differentiate into specific cell types [231,232]. This retention could be controlled using various compositions and ratios of polymers to adjust the network crosslinking degree, resulting in adjusted exosomes’ release time, targeted release, and biodegradability for specific clinical needs [233]. For example, polymers, such as chitosan and collagen are pro-osteogenic biomaterials, which could be fabricated to mimic ECM, leading the process to promote exosome-induced mineralization [232]. 

In the case of hard tissue reconstruction, such as craniomaxillofacial bones, scaffolds with mineral contents, with adequate mechanical and physical properties, are often needed. Various bio scaffolds have been developed to be incorporated by stem-cell-derived exosomes. Exosome-loaded hydroxyapatites were reported to promote calvaria bone regeneration, as shown by the enhanced counts of bone-forming fibroblasts, fibrocytes, osteoblast, osteocytes, and osteons, as well as the promoted expression of osteopontin [234,235]. In addition to hydroxyapatites, poly (lactic-*co*-glycolic acid) or PLGA [236], titanium [237,238,239], and β-tricalcium phosphate (β-TCP) [240] have been reported to be successfully applied in exosome-based bone tissue engineering, whether as a scaffold or an implant material [241]. These materials enhanced cellular migration toward the scaffold, then promoted cellular interaction and adhesion, leading to acceleration of osteogenic events [237]. Therefore, inorganic materials can increase the physical properties and mineral contents in the scaffolds to exert exosome bioactivities in inducing bone regeneration and mineralization, which can also be utilized for PDENs. Accordingly, it can be presumed that fabrication of PDEN-based bio scaffolds using ideal polymers in combination with ceramics could generate superior properties and clinical benefits for various applications. Although it is promising, the development of PDEN-based bio scaffold is still fully unexplored, which leads to difficult challenges ahead. However, based on the known properties, PDENs are certainly a promising biomolecule to construct bioactive scaffolds, as their anti-inflammatory, anti-bacterial, and antioxidant activities provide large benefits for regenerating tissues in clinical practice.

## 6. Summary and Future Perspectives

Despite numerous research on PDENs, current understandings on PDENs manufactures, biological mechanisms, and applications are still limited, and thus open wide areas for further research, development, and translational efforts. Moreover, the stability, quality, and standardization of PDENs during extraction would be of great challenges. Applications of plant-derived biomolecules are still raising questions, as it was stated that despite the fact that PDENs have comparable nucleic acid contents to mammalian cell-derived exosomes, their protein numbers are low, and possibly with different types of proteins. 

This would lead to the necessity of assuring the cross-species genetic regulation of PDENs. Validation on the functionality of the miRNAs inside PDENs in the human body or animal model is important, as a previous study reported that some bioinformatic approaches claimed that the miRNAs have no available targets in vivo [242]. Despite the abundance of studies on the biomedical benefits of PDENs, deep investigations on the cross-species regulation are needed to fully understand the bioactivity mechanism. Furthermore, a better understanding of PDENs in the future has the potential to usher in a new paradigm for natural medicine that employs compounds which are abundantly available, more effective, efficient, and have significantly fewer adverse effects than the currently accessible medications.

Physicochemical properties of PDENs make it a potential drug-delivery agent for various kinds of drugs, but it is noticeable that a deep understanding of drug-loading capacity and cellular transport of PDENs is essential. The ability of PDENs to promote the proliferative activity of cells and modulate the immune system by dampening inflammation is also observed, but substantial data on PDENs capacity to be applied in regenerative therapy are still uncovered. Moreover, current studies on PDENs properties, especially for the applications in tissue engineering, are almost non-existent. There are also limited studies on PDENs interaction with various biomaterials to resolve the remaining challenges in rapidly growing MSC-based therapies. Furthermore, it was found that studies on PDENs are mostly still at the in vitro level. This opens more possibilities and indispensable opportunities for comprehensive explorations, especially on their bioactivity, isolation, handling, and standardized mass manufacturing. Emerging technologies, such as biofunctionalization, bioink printing, and microfluidic system in the form of lab-on-a-chip may accelerate PDENs explorations in a more accurate and comprehensive way to construct proper strategies to encounter intricate diseases.

Combination of polymers and inorganic materials could also be a promising strategy for PDENs delivery to be applied in various biomedical treatments. An example is the application of PDENs loaded in periodontal membrane to alleviate periodontal disease and induce alveolar bone regeneration. It is well known that PDENs comprise various bioactive molecules, which apparently provide anti-bacterial, anti-inflammatory, and antioxidant properties. Antibacterial properties of PDENs could eliminate etiological periodontopathogens, restrain cellular death, and regulate massive inflammatory mediators. When loaded with PDENs, hybridization of bioceramics and polymers could result in osteoconductive scaffolds with controlled release of PDENs. The construct will induce resident cellular adhesion and initiate bone formation, along with promoted fibroblasts proliferation to heal gingival destruction. 

## Figures and Tables

**Figure 1 biomedicines-11-01053-f001:**
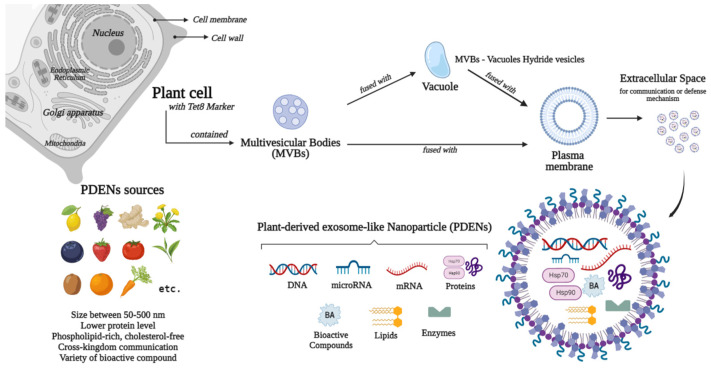
Sources, biogenesis, and contents of plant-derived exosome-like nanoparticles (PDENs).

**Figure 2 biomedicines-11-01053-f002:**
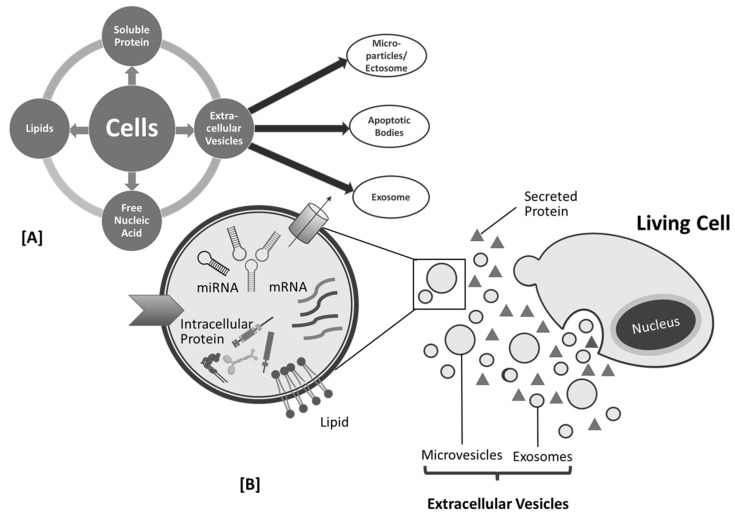
Cell products (**A**) and biogenesis of mesenchymal stem cell (MSC)-derived exosome (**B**).

**Table 1 biomedicines-11-01053-t001:** Metabolites found in PDENs from several sources.

Sources of PDENs	Compounds	Remarks
Ginger	6-gingerol, 8-gingerol, 10-gingerol, and 6-shogaol	[88,108]
Grapefruit	Naringenin	[65]
Lemon	Citrate, vitamin C, and galacturonic acid-enriched pectin-type polysaccharide	[66,110]
Strawberry	Vitamin C	[88]
Tea flower	Epigallo-catechin gallate, epicatechin gallate, epicatechin, vitexin, myricetin-3-*O*-rhamnoside, kaempferol-3-*O*-galactoside, and myricetin	[111]
Oat	Fiber β-glucan	[112]
Broccoli	Sulforaphane	[112]

**Table 3 biomedicines-11-01053-t003:** Cells used in the experiments on the capacity of the vesicles to act as a potential biomedical therapeutical agent.

Type of Cells	References
Human keratinocyte line (Ha-CaT), human skin fibroblasts (HSFs), human embryonic kidney cells (HEK293T)	[188,198]
Fibrotic cell line (LX2), hepatocellular cell line (HEP40)	[189]
Cancer cells (MCF-7, HeLa, and N2A), human umbilical vein endothelial cells (HUVEC), primary fibroblasts cells	[191,197]
Human colorectal cancer cell line (DLD-1), human neuroblastoma cell line (NB19), human leukemia cell line (K562), ASF-4-1 cells, RD cells	[192]
Human alveolar basal epithelial cells (A549)	[182,185]
Monocytic U937 cell lines	[185]
Human hepatoblastoma cells (HepG2)	[193,194]
SMMC-7721 cells, Hep3B cells, normal hepatocyte cell line (LO-2)	[194]
Adipose-derived mesenchymal stem cells (ADMSCs)	[88]
Human stabilized endothelial cell line (EA.hy926)	[195]
Human telomerase immortalized keratinocytes (TIGKs)	[186]
Human epithelial colorectal adenocarcinoma cells (Caco-2)	[101,202]
Human chronic myeloid leukemia cell line (LAMA84), human colorectal adenocarcinoma cell line (SW480), human bone marrow-derived stromal cell line (HS5)	[182]
Human fetal lung fibroblast (MRC-5), human lung adenocarcinoma (A549), human acute monocytic leukemia (THP-1)	[198]
KB cells	[199]
Mouse hippocampal neuronal cells (HT22)	[190]
Mouse C57BL/6 lung carcinoma cells (LLC1), monkey kidney Vero E6 cell	[185]
Mouse bone marrow-derived macrophages (BMDMs)	[196]
C57BL/6 murine colon adenocarcinoma cells (MC-38)	[101]
Murine RAW 264.7 cells	[200]

**Table 4 biomedicines-11-01053-t004:** Summary of review articles on PDENs in biomedical approaches.

Potential Target Tissue or Disease	Prospects	Challenges and Issues	Refs.
Periodontitis	PDENs could steadily carry drugs for oral mucosal delivery to modulate oral immunity to periodontopathogen	PDENs could only load small amounts of drugs, with an unclear mechanism of cellular uptake as it may vary per extraction batch	[69]
Colitis, tumor, liver diseases, skin diseases, periodontitis	PDENs could act as a drug-delivery system to deliver RNAs and lipids for the inhibition of inflammation genes, as well as bacterial and tumor growth	No clarity on quality control and evaluation systems, stability, biomarker confirmation, and biochemical characterization	[204]
Unspecified	Easier large-scale mass production for their large biodiversity with minimal cytotoxicity for drug-delivery system	Elusive internalization mechanisms and unclarity of specific receptors and ligands for PDENs. Biosafety and toxicity of genetic transfer are unclear	[205]
Intestine	PDEN-derived miRNAs could modulate gut microbiome, intestinal permeability, and mucosal immunity	High variability of results in studies, due to the lack of consensus regarding PDEN-derived miRNAs	[206]
Inflammatory bowel disease, liver disease, cancer	PDENs could mediate interspecies communications to exert their antioxidant, anti-inflammatory, and regenerative activities	Low amount and different kinds of PDEN-derived proteins compared to MSC-derived exosomes	[207]
Colitis	PDENs could transfer exogenous drugs or endogenous cargo to epithelial and bacterial cells for their stability in intestinal fluid	Standardization on mass producing and purification techniques	[72]
Unspecified	PDENs stability in the digestive system make it a promising functional food to alleviate inflammation	Instability during isolation and processing with unclear proteomic profiling	[21]

Abbreviations: PDENs = plant-derived exosome-like nanoparticles; miRNA = micro-RNA; MSC = mesenchymal stem cell.

## Data Availability

Not applicable.

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
