# Peer review of "Plant-Derived Exosome-like Nanoparticles for Biomedical Applications and Regenerative Therapy"

_biomedicines, 2023, doi:10.3390/biomedicines11041053_

Round 1
Reviewer 1 Report
This manuscript claims to be a comprehensive review of plant-derived extracellular vesicles (PDEVs) in biomedical applications and regenerative therapy. EVs are produced by almost all cells from the living organisms, from the single-celled organisms to higher mammals. Some think that the EV may communicate across Kingdoms and modify biological activity in recipient organisms. The experimental evidence supported that the EV derived from the plants can be taken into animal cell and subsequently modify cell behavior. The current manuscript summarizes the latest evidence supporting this concept. The topic in interesting, and the plant-derived extracellular vesicles have great research value and important potential application. There are several comments to help the authors improve their manuscript.
1. The language should be modified by a native-English speaker or language-edit service.
2. This review is mainly focused on the exosome-like nanoparticles derived from plant. The context of the MSCs derived nano-sized vesicular should be removed from the abstract section.
3. There is logical confusion in the abstract section. Please re-organized this part.
4. Although the nano-sized vesicular are mainly derived from MSCs, there are still many other ways or source, such as the liposomes technology. A comprehensive summarizes of the source of the EV used in clinical should be provided before claimed that “still problematic when it comes into large scale clinical applications”
5. In the figure 1, the plant cells should have the structure of cell wall.
6. The authors claimed that “cross-species genetic regulation of PDEN”. That is a very important statement which might affect the current research field and clinical application. Please provide citations and lay it out in detail.
7. This manuscript is mainly focused on the potential value and usage of PDEVs in biomedical applications and regenerative therapy. But it seems most of the contents are involved in the produce and secrete of EVs, but not the applications of the EVs.
8. The literatures should be cited point to points. The individual literature should be added to the corresponding knowledge point, instead of all listed at the end of the sentence. For example, “Exosome functions in cellular communication, nutrients, bioactive compounds delivery, and cellular immunity [19–22]”, “but their application is limited due to ethical issues and the danger of unlimited and uncontrolled cells division [5–15]”
9. The Figures 1 and 2 do not provide much information. It should be removed from the main text or instead by Information-rich content. For example, how the plant cell produces and secrete/release the EVs? A comprehensive summarizes of the latest knowledge of the mechanism of EVs/exosome production and release should be used to instead the current Figure 1.
10. PDEN, PDENs, please keep consistent.
Author Response
Attached is our consolidated comments.

Reviewer 2 Report
The article presents the current state of research on plant-derived exosome-like nanoparticle (PDEN) in the idea of organizing the research results of recent years in several feasible implementation areas.
Apparently, their regenerative, anti-inflammatory and antimicrobial effects appear as most feasible directions and biomedical applications too.
Considering the rather large number of PDEN materials published in the recent years, in the same idea of better valorization of scientific results nowadays (mostly from in vitro studies), additions are needed regarding: methods of isolation from vegetal raw materials, and methods of chemical preparation/synthesis in the laboratory; data regarding their degradation and sensitivity in contact with human living tissues; data regarding chemical analysis and standardization of these biopreparations; the advantages and disadvantages of a cholesterol-free vehicle (PDEN type) compared to those containing cholesterol, in human tissue interaction purpose.
Author Response
The article presents the current state of research on plant-derived exosome-like nanoparticle (PDEN) in the idea of organizing the research results of recent years in several feasible implementation areas.
Apparently, their regenerative, anti-inflammatory and antimicrobial effects appear as most feasible directions and biomedical applications too.
Considering the rather large number of PDEN materials published in the recent years, in the same idea of better valorization of scientific results nowadays (mostly from in vitro studies), additions are needed regarding: methods of isolation from vegetal raw materials, and methods of chemical preparation/synthesis in the laboratory; data regarding their degradation and sensitivity in contact with human living tissues; data regarding chemical analysis and standardization of these biopreparations; the advantages and disadvantages of a cholesterol-free vehicle (PDEN type) compared to those containing cholesterol, in human tissue interaction purpose.
We have accommodated the suggestions and views of the reviewers in the revised text. However, for more detailed information regarding methods of isolation from vegetal raw materials, and methods of chemical preparation/synthesis in the laboratory; data regarding their degradation and sensitivity in contact with human living tissues; data regarding chemical analysis and standardization of these biopreparations; the advantages and disadvantages of a cholesterol-free vehicle (PDEN type) compared to those containing cholesterol, in human tissue interaction purpose, it is still on going works for us at the lab.

Reviewer 3 Report
This review recapitulates the potential of PDEN for human health encouraging their use in biomedicine, similarly to many other recent studies underlining their therapeutical importance.
-the review deals quite well the proposed topic although there are several typos and inaccuaracies in the references (some are duplicated).
- in addition, it is not mentioned the recent finding of extracellular vesicles secreted by Salvia hairy roots, showing antitumor activity. In fact, hairy roots of medicinal plants could be a precious source of therapeutical extracellular vesicles du to their natural content in pharmaceutically active compounds (as shown for examples in Vaccaro et al., 2020 and others). So, hairy roots shows a tremendous potential as source of PDEN.
-Resolution of figure 1 should be improved;
-in table 1 and table 4 the font should not be "justified".
Author Response
This review recapitulates the potential of PDEN for human health encouraging their use in biomedicine, similarly to many other recent studies underlining their therapeutical importance.
-the review deals quite well the proposed topic although there are several typos and inaccuaracies in the references (some are duplicated) – We have already checked and revised it.
- in addition, it is not mentioned the recent finding of extracellular vesicles secreted by Salvia hairy roots, showing antitumor activity. In fact, hairy roots of medicinal plants could be a precious source of therapeutical extracellular vesicles due to their natural content in pharmaceutically active compounds (as shown for examples in Vaccaro et al., 2020 and others). So, hairy roots shows a tremendous potential as source of PDEN – We have already accommodated this views in the revised texts.
-Resolution of figure 1 should be improved – Revision has been done.
-In table 1 and table 4 the font should not be "justified" – Revision has been done.

Round 2
Reviewer 1 Report
Thanks for the author's reply. All of my comments have been answered. The current version is suitlable for publication in this journal.
Author Response
From the reviewer:
"Thanks for the author's reply. All of my comments have been answered. The current version is suitlable for publication in this journal. "
Author comment: Thank you very much for the comments. We hope this paper will be beneficial.
Reviewer 2 Report
The authors brought new information, very useful for the readers, at the same time they responded to the request for major revision according to the suggestion. Therefore, the publication of the revised article is accepted.
Author Response
Reviewer comment:
"The authors brought new information, very useful for the readers, at the same time they responded to the request for major revision according to the suggestion. Therefore, the publication of the revised article is accepted."
Author comment:
Thank you for the review. We hope this will be beneficial.